# Trustworthy Retrosynthesis: Eliminating Hallucinations with a Diverse Ensemble of Reaction Scorers

## Abstract

Retrosynthesis is one of the domains transformed by the rise of generative models, and it is one where the problem of nonsensical or erroneous outputs (hallucinations) is particularly insidious: reliable assessment of synthetic plans is time-consuming, with automatic methods lacking. In this work, we present RetroTrim, a retrosynthesis system that successfully avoids nonsensical plans on a set of challenging drug-like targets. Compared to common baselines in the field, our system is not only the sole method that succeeds in filtering out hallucinated reactions, but it also results in the highest number of high-quality paths overall. The key insight behind RetroTrim is the combination of diverse reaction scoring strategies, based on machine learning models and existing chemical databases. We show that our scoring strategies capture different classes of hallucinations by analyzing them on a dataset of labeled retrosynthetic intermediates. To measure the performance of retrosynthesis systems, we propose a novel evaluation protocol for reactions and synthetic paths based on a structured review by expert chemists. Using this protocol, we compare systems on a set of 32 novel targets, curated to reflect recent trends in drug structures. While the insights behind our methodology are broadly applicable to retrosynthesis, our focus is on targets in the drug-like domain. By releasing our benchmark targets and the details of our evaluation protocol, we hope to inspire further research into reliable retrosynthesis.

## 1 Introduction

The advent of deep generative modeling has transformed a broad range of domains, and resulted in impressive applications such as photorealistic image synthesis, code generation, and automated theorem proving (Rombach et al., 2022; Li et al., 2022; Yang et al., 2023). The undisputed efficacy of generative models is nonetheless hindered by the possibility of factually wrong or outright nonsensical output, often called "hallucinations" (Sahoo et al., 2024). In some applications, such as automated theorem proving, hallucinations can be curtailed with formal verification of output. In others, they can only be mitigated through partial and often imprecise verification.

Retrosynthesis, the task of constructing synthetic routes — sequences of chemical reactions that lead to a desired target molecule from simpler precursors — is another domain that has undergone significant developments with the rise of generative modeling (Coley et al., 2018)). Modern systems typically generate complete synthetic routes by iteratively applying predictions from a single-step retrosynthesis (SSR) model within a graph-based search algorithm. When reactions have no known chemical precedent or display features such as unstable reagents, they jeopardize the validity of the synthetic path they are part of. Generative SSR models inherit the limitations of generative modeling: the reactions they propose are frequently hallucinated, that is, chemically unsound (Torren-Peraire et al., 2024). An example of a hallucinated reaction is shown in Figure 1

In this work, we propose RetroTrim, a retrosynthesis system designed to tackle the issue of hallucinated reactions by pairing a performant SSR generator with an ensemble of complementary reaction scorers as a plausibility filter. We show that RetroTrim is the only among common retrosynthetis solutions that successfully avoids *all* hallucinated reactions in proposed synthetic plans for a set of

Figure 1: An example of grossly incorrect (hallucinated) reaction generated by a Single-Step Retrosynthesis model. A PhD-level chemist recognizes that the only reasonable atom mapping between the substrates and the product is one where the reaction center is an ortho-amino benzoate converting into a triazole (highlighted in yellow). It does not belong to any commonly known reaction class, and further investigation involving extensive searches of synthetic databases yields no examples that would inform what reagents and conditions could induce such a reaction. Executing this transformation would be impractical and require the development of a novel synthetic methodology, which typically entails a multi-month research program.

unpublished challenging drug-like targets. Moreover, RetroTrim does so while leading in the number of targets for which a synthetic plan without issues is found.

Our approach in developing RetroTrim is based on the principle of ensemble learning (Hansen & Salamon, 1990): combining diverse scorers with distinct error patterns yields a more robust and accurate assessment of reaction plausibility. The ensemble scorer (MetaScorer) functions as an external validator of the generative SSR model. It prunes all reactions below a given score threshold from the synthesis tree, curtailing expansion of hallucinated nodes. For an overview see Fig. 2. The three scorers which make up the plausibility filter are as follows:

1. Reaction Prior (RP): A Transformer-based architecture (Vaswani et al., 2017) whose scoring function is designed to mimic the considerations chemists take into account when evaluating reaction plausibility. Its training and scoring method lends itself to discovering a broad spectrum of hallucinations.

2. Reaction Graph Plausibility (RGP): A graph model trained to distinguish positive/feasible reactions from synthetically generated negatives. Negative reactions are generated by applying reaction templates at random in both the forward and retro- direction. The forward negatives are designed to increase fidelity in discriminating selectivity problems, while retro- negatives lead of a broader spectrum of incompatibilities between functional groups in reactants.

3. Reaction Retrieval Score (RRS): A mechanism that assesses the similarity of proposed reactions to known experimental precedents in reaction databases. It is designed to catch hallucinations of the more gross kind: reactions with no precedents and significant mismatches between the target and reactants.

We compare performances of RP, RGP and RRS on a dataset of retrosynthetic intermediates. We show that each scorer excels at filtering different kinds of hallucinations, and that the MetaScorer further improves performance in most areas.

In identifying hallucinations of retrosynthesis systems, common automated metrics can at best serve as surrogates: algorithmic procedures cannot model the whole concept of chemical plausibility, and exact matches to literature are rare in practice for complex targets. The only way to arrive at a verdict on a synthetic route in practice is to propose it to one or more expert chemists. If the route contains reactions for which no known synthetic methodology exists, such a route will be rejected without the need for experimental verification.

For this reason, to measure the performance of both retrosynthesis systems as well as reaction scorers, we employ a novel reaction validation protocol in which PhD-level chemists assess each reaction in a given synthetic plan according to a predefined labeling schema. Reactions are sorted into seven categories depending on the kind of issue they raise: *Magic*, *Selectivity*, *Functional group incompatibility*, *Reactivity*, *One pot*, *Unstable*, or *Reactants mismatch*; and according to the severity of said issue: *Worthwhile*, *Rather not*, and *Nonsense*. Reactions which present no issues are given the

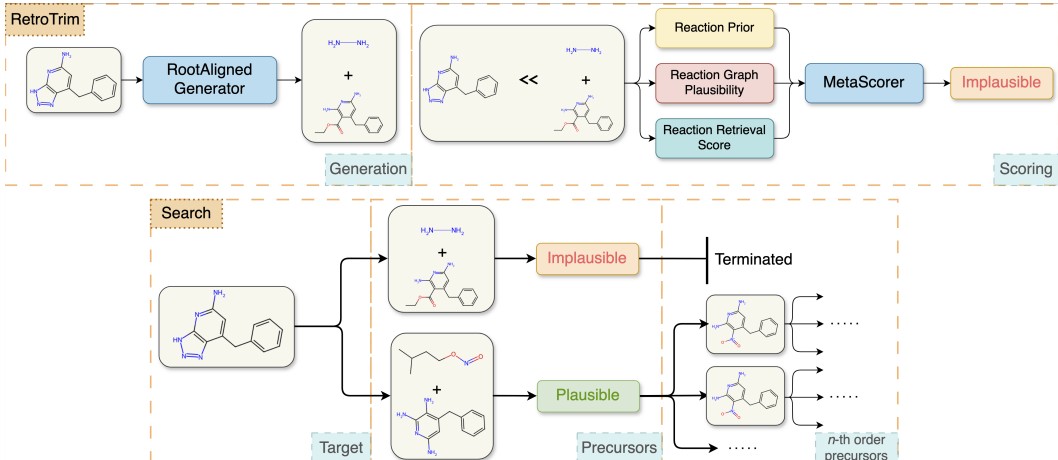

Figure 2: Visualization of **RetroTrim** (above) and **retrosynthetic search with plausibility filtering** (below). RetroTrim encompasses a generator, which proposes precursor molecules for a given target, and a scorer, which evaluates the plausibility of the generated reaction. In the search process, plausible precursors are expanded further, until we arrive at commercially available starting materials. Implausible reactions terminate the search branch. The search concludes when a complete synthetic route from commercially-available starting materials to the target molecule is found.

special labels *No problem* and *Safe bet*, respectively. Under this categorization, *Nonsense* corresponds to hallucinated reactions, while *Worthwhile* and *Rather not* reactions contain mild or moderate issues which make them undesirable or unpredictable. *The overall confidence in a synthetic pathway is determined by the reaction with the highest issue severity within it.*

All retrosynthesis systems are evaluated on a set of thirty-two challenging drug-like targets (see Appendix D). These targets were defined so as to match common structural features present in modern drugs while not being close analogs of any structure found in synthetic literature. This ensures no data leakage occurred in-between training and inference of compared models, while still being representative of the medicinal chemistry domain.

Our main contributions are as follows:

- We propose RetroTrim which achieves state-of-the-art accuracy, rejecting as the only method all hallucinated reactions for a set of challenging targets while at the same time finding flawless synthetic routes for the largest number of them.

- We demonstrate that the three scorers used in RetroTrim as reaction filters display different strengths across potential issues with reactions, and furthermore that they are complementary: combining them leads to the best-performing system.

- We outline a reaction labeling protocol that recognizes seven kinds of possible issues at three levels of severity, to be used by expert chemists for granular evaluation of both reactions and the synthetic routes composed of them.

- We release a set of 32 unpublished drug-like targets, designed as a challenging test set for retrosynthesis systems.

## 2 RELATED WORK

To contextualize our proposed method, this section examines the workflow in automated retrosynthesis, covering both the generation of reactions by Single-Step Retrosynthesis (SSR) models and techniques used for their validation.

## 2.1 Multi-Step Retrosynthesis

Multi-step retrosynthesis constructs complete synthetic routes by iteratively applying single-step retrosynthesis (SSR) predictions. Classical search strategies, such as Monte Carlo tree search (Segler et al., 2018) and A* search (Retro*) (Chen et al., 2020), use SSR models to expand compounds during generation of synthetic pathways.

SSR models can be categorized into template-based, template-free, and semi-template-based approaches. Template-based methods (e.g., RetroSim (Coley et al., 2017), NeuralSym (Segler & Waller, 2017), GLN (Dai et al., 2019a)) use predefined reaction templates, ensuring interpretability, but their coverage is limited and their construction involves trade-offs that can lead to invalid template application. Template-free models (e.g., sequence-to-sequence Transformers (Zhong et al., 2022), GNNs (Liu et al., 2017; Karpov et al., 2019; Sacha et al., 2021)) learn chemical rules directly from data, offering scalability and the ability to generalize to novel reactions, but with little guarantee as to the validity of output. Semi-template-based approaches (e.g., GraphRetro (Somnath et al., 2021), RetroXpert (Yan et al., 2020)) combine templates with learned representations, inheriting both the advantages and limitations of both approaches. Regardless of their category, SSR models are all prone to erroneous output (Coley et al., 2018).

Evaluating single-step retrosynthesis (SSR) models remains challenging. Traditional metrics, such as top-k accuracy, measure whether the correct reactants appear among the top predicted candidates, but they provide limited insight into chemical plausibility and alternative valid reactions.

To address this, round-trip accuracy (Schwaller et al., 2020) is commonly used. This metric evaluates a predicted reaction by passing the predicted reactants through a forward reaction model and checking if the original product is recovered. Round-trip accuracy is considered a better proxy for chemical validity than top-k accuracy.

However, it remains unclear how well round-trip accuracy reflects the actual plausibility of the predicted reactions. There is currently no exhaustive data quantifying how many predictions that pass or fail the round-trip check correspond to chemically plausible reactions.

Evaluating multi-step retrosynthetic routes is typically limited to basic pathway properties, such as the number of steps, route length, branching factor, or overall synthesizability (Maziarz et al., 2025). These metrics provide little insight into the actual correctness or chemical plausibility of the complete synthetic route.

Evaluation of multi-step routes by expert chemists provides a more trustworthy viability assessment. To our knowledge, the only work containing human evaluation of routes powered by generative models is Segler et al. (2018), which conducts a double-blind A/B test comparing routes existing in literature and those produced by their model. Their results showed that for the first time, computer-generated routes could be on-par or preferred to those found in the literature by chemists. While a significant result in its own right, their test only recorded preference between paths as opposed to measuring the rate of significant system errors. It also spanned only 9 randomly chosen targets. In contrast, we conduct fine-grained error-analysis of routes proposed by 4 modern retrosynthesis systems, as well as 6 variants of our own system, including the full RetroTrim and simpler baselines. Moreover, we do so on a set of 32 targets chosen for their challenging nature.

## 2.2 Mid-Search Reaction Validation

The plausibility of multi-step retrosynthetic routes has drawn increasing attention from researchers, leading to the development of methods aimed at improving pathway validity. A commonly used approach is to employ a plausibility model as a filter. Such models are typically trained as classifiers to distinguish correct reactions from artificially generated negative examples, which are often created by perturbing positive reactions, for example by randomly modifying substrates or swapping the original product with a chemically similar one (Segler et al., 2018; Genheden et al., 2020a). A forward model (predicting products based on substrates) trained only on positive data was used in the same way in IBM RXN (Schwaller et al., 2020).

Another strategy is to use the likelihood of a generated reaction as a scoring function during search. This can be computed from template-free models, such as the Molecular Transformer (Schwaller et al., 2019) architecture, where the model's confidence, derived from output token probabilities, has

been shown to correlate with reaction correctness, or from template-based models using softmax scores, as in RetroFallback (Tripp et al., 2023). In order to improve the quality of predicted reactions, RetroGFN (Gaiński et al., 2024) was trained using feedback from a forward model and round-trip accuracy. RetroChimera (Maziarz et al., 2024) takes a complementary approach by selecting the most plausible predictions from an ensemble of diverse reaction generators, effectively combining their strengths to increase overall accuracy.

Finally, evidence-based validation via retrieval grounds model predictions in established chemical knowledge, mirroring a chemist's workflow of searching for literature precedents. Retrieval-augmented methods, such as the Retrieval-Augmented RetroBridge (RARB) framework, retrieve similar molecules from a database to guide the generation of reactants (Qiao et al., 2025). These approaches help ensure that generated reactions are consistent with known chemistry, further improving the plausibility of multi-step routes. In the context of these works, RetroTrim is the first to use retrieved reactions to filter predictions of a reaction generation.

To date, none of these approaches have been extensively validated through human expert evaluation.

## 3 METHODS

RetroTrim is a combination of an SSR reaction generator and a reaction scorer. The key innovation in RetroTrim is the design of the scorer, which is built around a central, best-performing scorer called Reaction Prior, supported by two additional scorers: Reaction Graph Plausibility scorer and Reference Reaction Scorer. The additional scorers are designed to compliment Reaction Prior by targeting specific error types. Together, they aggregate into the MetaScorer which provides the final, robust assessment of reaction correctness.

### 3.1 REACTION PRIOR

The Reaction Prior (RP) is a novel method inspired by how experienced chemist reason about reactions: considering the reaction globally, assessing the reaction center, and comparing it to alternative reactions that could occur. The RP score $S_{final}$ is thus a weighted combination of three components: $S_{final} = S_{RP}^{\alpha} \cdot S_{Regio}^{\beta} \cdot S_{RC}^{\gamma}$. Here, $S_{RP}$ is the Reaction Prior (global) Score, $S_{RC}$ is the Reaction Center Score, and $S_{Regio}$ is the Regioselectivity Score, with $\alpha$, $\beta$, and $\gamma$ serving as weighting factors. These weights can be tuned to optimize a metric of interest. In practice, we find $\alpha = 1$, $\beta = 1.5$ and $\gamma = 2.5$ to be reasonable default values for drug-like targets. RP is implemented as an autoregressive, encoder-decoder BART (Lewis et al., 2019a) architecture, where substrates and products are processed by the decoder, trained for next-token prediction by minimizing cross-entropy loss.

**Reaction Prior Score** ($S_{RP}$)  This score reflects the overall feasibility of the reaction. It is proportional to the log probability assigned by the model to the reaction sequence. Analogously to language models, which assign higher probabilities to sequences close to the training set, this score acts as a measure of reaction similarity to the dataset on which the model is trained on. In this respect, it mimics the way in which a chemist would look for precedent for the overall transformation. We normalize the log probability by the square root of the total number of tokens ($T$), which we find to maximize predictive performance in practice: $S_{RP} = \frac{1}{\sqrt{T}} \log P(\text{reaction})$.

**Reaction Center Score** ($S_{RC}$)  When evaluating reactions, chemists pay special attention to the sites where changes in connectivity occur. The $S_{RC}$ score analogously evaluates the model's confidence in the identified reactive sites. It is proportional to the sum of log probabilities of tokens representing atoms in the reaction's center. Unlike $S_{RP}$, here we normalize by the number of such tokens ($T_{RC}$), similarly guided by empirical calibration: $S_{RC} = \frac{1}{T_{RC}} \sum_{i \in \text{reaction center}} \log P(\text{token}_i)$.

**Regioselectivity Score** ($S_{Regio}$)  This component quantifies reaction site specificity. It is calculated by comparing the probability of the reaction at the given reaction site ($P_{\text{desired}}$) to the summed probabilities of all sites where the reaction might occur ($P_{\text{undesired}}$): $S_{Regio} = \log\left(\frac{P_{\text{desired}}}{P_{\text{undesired}} + \epsilon}\right)$, where $\epsilon$ is a small constant to prevent division by zero. This score component reflects the tendency for

chemists to evaluate whether the particular site where a reaction occurs is preferred compared to potential alternatives.

## 3.2 REACTION GRAPH PLAUSIBILITY

A Graph Attention Network (GAT) (Veličković et al., 2017) is trained to differentiate chemically valid reactions from implausible ones. Training uses reaction datasets for positive examples and synthetic negative examples generated through forward and two-step backward template applications. This approach is similar to those proposed by (Segler et al., 2018; Genheden et al., 2020a), but it uses a graph neural network instead of a feedforward network with fingerprint inputs, and employs more sophisticated artificial negative reactions, generated not only by applying random templates in the forward direction but also through retro-synthetic random template applications, which increases the diversity of types of generated incorrect reactions. Details of GAT featurization are described in Appendix B.

## 3.3 REFERENCE REACTION SCORER

We developed a structured retrieval pipeline that extracts chemical precedent information through a two-tiered reaction clustering procedure based on bond change patterns. First *Coarse-grained clustering* extracts connected components of the reaction center and applies atom mapping to identify the underlying transformation pattern. Reactions belong to the same cluster if their transformation patterns are identical. Then *Fine-grained clustering* extends the coarse-grained approach by incorporating chemically significant substructures — aromatic systems and conjugated double bonds - into the cluster classification.

Our Reference Reaction Retrieval Scorer (RRS) quantifies reaction plausibility through a logarithmic transformation of the unique reference reaction count, where we sum the number of coarse-grained and fine-grained references of a candidate reaction:

$$p(\texttt{reaction}) = \log(n_{\text{ref}}(\texttt{reaction}) + 1) \tag{1}$$

where $n_{\text{ref}}(\texttt{reaction})$ represents the unique number of reference reactions in the coarse-grained and fine-grained clusters containing reaction.

## 3.4 METASCORER AGGREGATION

To improve reaction filtering, our MetaScorer integrates scores from Reaction Prior ($s_{\text{RP}}$), Reaction Graph Plausibility ($s_{\text{RGP}}$), and empirical precedents ($n_{\text{ref}} > 0$) retrieved via the pipeline in Sec. 3.3. This hybrid approach mitigates the weaknesses of purely data-driven or precedent-based methods. The continuous score is described by equation $s_{\text{META}} = \max(s_{\text{RGP}}, s_{\text{RP}})$ if $n_{\text{ref}} > 0$ (0 otherwise).

For binary classification tasks and search, reactions are filtered using predefined thresholds, which can be selected through grid search to balance precision and recall:

$$s_{\text{META}} = \begin{cases} 1 & \text{if } s_{\text{RGP}} > \texttt{thr}_{\text{RGP}} \text{ and } s_{\text{RP}} > \texttt{thr}_{\text{RP}} \text{ and } n_{\text{ref}} > 0 \\ 0 & \text{otherwise} \end{cases} \tag{2}$$

By synthesizing diverse evidence types MetaScorer enables more reliable reaction filtering for multi-step synthesis planning. This integrated approach mitigates individual weaknesses of purely data-driven or precedent-based methods, yielding improved performance.

## 3.5 GENERATOR AND INTEGRATION WITH SEARCH (RETRO*)

For the generator part of RetroTrim, we use the encoder-decoder BART architecture (Lewis et al., 2019b) trained on root-aligned SMILES (Zhong et al., 2022), where the product (target) is processed by the encoder, and the substrates are generated by the decoder. We call this generator RootAligned. The calibrated MetaScorer is used during multi-step retrosynthesis search to improve the quality of the pathways predicted by the BART generator. We integrate the scorer into the Retro* (Chen et al., 2020) search framework as a reaction filtering mechanism. Reactions are pruned from the search tree if $s_{\text{META}}$ defined in 2 is equal to 0.

## 4 HUMAN EVALUATION

We curated a dataset of over 4,500 reactions generated by our SSR models. Each reaction was evaluated and labeled by PhD-level chemists into one of the expert-defined categories, creating the first comprehensive dataset of its kind. This datased provides a way to evaluate the error patterns of our reaction plausibility scorers. A subset of 500 reactions from this dataset will be released as a benchmark for the community.

**Reaction Evaluation Protocol** was designed to systematically evaluate predicted reactions based on expert-defined heuristics. Reactions were rated using a four-point confidence scale: *Nonsense*, *Rather not*, *Worthwhile*, and *Safe bet*. *Safe bet* reactions are considered fully reliable; *Worthwhile* reactions remain plausible but carry a moderate risk of failure; *Rather not* reactions are associated with a high risk of major difficulties; and *Nonsense* reactions are effectively infeasible, i.e. hallucinated. For the system to be reliable, valid pathways should consist primarily of *Safe bets* reactions. The presence of *Nonsense* reactions effectively invalidates a pathway, while the presence of a *Rather Not* reaction may still be acceptable in target-oriented synthesis when no alternatives exist. Reactions which aren't a *Safe bet* receive an additional label specifying the cause of their incorrectness, chosen from: *Reactants mismatch*, *Unstable*, *Magic*, *One pot*, *Reactivity*, *Functional group incompatibility*, and *Selectivity*. These error categories correlate with confidence levels to varying degrees: for example, *Magic* errors almost always map to *Nonsense*, while *Selectivity* issues more often correspond to *Worthwhile* or *Rather Not*. Otherwise, a reaction is assigned a *No Problem* label. A detailed description of the evaluation framework is provided in Appendix C.

## 5 EXPERIMENTS

### 5.1 DATASET

All of our generators and scorers are based on the proprietary Pistachio (2024Q3 release) (Mayfield et al., 2017) dataset, either used as training data (RootAligned, RP, RGP), or as a source of reference reactions (RRS). Pistachio offers substantial advantages over the commonly used USPTO-50K (Schneider et al., 2016) and USPTO-FULL (Dai et al., 2019b) datasets - it features enhanced curation, resulting in higher data quality and more comprehensive coverage of chemical reaction space. For training, we preprocess the dataset through a multi-step filtering pipeline that removes duplicate reactions, reactions from unrecognized reaction classes, entries with invalid SMILES, unmapped reactions, and reactions deemed unrelated to drug-like compound synthesis (molecules with >100 atoms, "separation" reaction classes), retaining approximately 4 million reactions.

### 5.2 PATH-LEVEL PLAUSIBILITY EVALUATION

We compared the ability of each individual scorer (RP, RGP, and RRS) as well as the MetaScorer at filtering out implausible reactions in multi-step retrosynthesis. All scorer variants were used together with our BART-based RootAligned generator.

The thresholds for filtering implausible reactions were maximized under the constraint that the resulting system finds paths for at least 90% of targets. In practice, this corresponded to a precision value of 0.8 on the reaction dataset described in 4. We targeted both a relatively strict threshold (to keep precision high) and broad coverage in terms of targets for which routes are found. Importantly, these thresholds were not tuned to optimize benchmark performance. As baselines, we report results for RootAligned without scoring, RootAligned with a forward reaction scorer, and LocalRetro (Chen & Jung, 2021). The forward scorer is implemented in the same BART architecture as RootAligned, except it is trained to predict products based on substrates. For the search, we used the widely adopted Retro* algorithm based on the implementation from (Maziarz et al., 2025) with the expansion limit set to 500. For all systems, we used the same starting material database, eMolecules.

Additionally, we compared against three publicly accessible retrosynthesis systems: AiZynthFinder (Genheden et al., 2020b), IBM RXN (IBM, 2025), and RetroChimera (Maziarz et al., 2024). AiZynthFinder was used in its default configuration from the official repository, which includes a template generator with a filtering model trained to distinguish valid reactions from artificially generated negatives. The only modification we make is an increased time limit of 15 minutes to better match

the runtime of other systems. IBM RXN, which makes use of a forward model in its search, was accessed through its free web application (IBM, 2025). RetroChimera was queried via the Azure Foundry multi-step retrosynthesis endpoint (Microsoft, 2025). Due to a low per-call timeout, we repeated the queries multiple times for each target; thanks to prediction caching, this resulted in an effective 15-minute search time. Like RetroTrim, RetroChimera does not employ an explicit reaction scorer, instead it aims to enforce plausibility by selecting the top-ranked reactions from an ensemble of generators.

We evaluated the top-1 synthesis paths generated by all systems for 32 selected targets (listed in D). Each reaction in the generated paths was manually evaluated by expert chemists according to the evaluation protocol described in 4. Each path was assigned a four-tier confidence score (*Safe bet*, *Worthwhile*, *Rather not*, *Nonsense*), determined by the lowest-scoring reaction in the path. This conservative scoring reflects the intuition that a single implausible step can invalidate an otherwise promising synthesis. Increasing the proportion of *Safe bet*s, while eliminating *Nonsense* and reducing *Rather Not* paths is the goal of all retrosynthesis systems.

## 5.3 REACTION-LEVEL PLAUSIBILITY PREDICTION

Additionally, we compare the performance of individual scorers (RP, RGP and RRS) and the MetaScorer on individual reactions with ground truth labels established through expert chemist evaluations described in Section 4.

Model performance was assessed using precision-recall (PR) and receiver operating characteristic (ROC) curves, with area under the curve metrics (PR-AUC and ROC-AUC) reported for each method. Reactions with confidence rating *Safe Bet* were treated as positive examples. *Worthwhile* reactions were excluded from the test set as they represent borderline cases where chemist confidence is uncertain, making them neither clearly positive nor negative examples for evaluation purposes. All others (*Rather Not* and *Nonesense*) were labeled as negatives. We also conducted additional analysis across each failure category, reporting individual ROC-AUC and PR-AUC scores, as well as false positive counts.

To evaluate model complementarity, we analyzed the overlap in false positive predictions across individual scorers, calculated as:

$$\text{overlap} = \frac{\left| \bigcap_{\text{scorer} \in \{\texttt{RGP},\texttt{RP},\texttt{RRS}\}} \text{FP}_{\text{scorer}} \right|}{\min_{\text{scorer} \in \{\texttt{RGP},\texttt{RP},\texttt{RRS}\}} \left| \text{FP}_{\text{scorer}} \right|}, \tag{3}$$

where FP is a set of false positives produced by a given scorer.

## 6 RESULTS

### 6.1 PATH-LEVEL PLAUSIBILITY EVALUATION

Pathway correctness comparison is presented in the figure 3. AiZynthFinder demonstrates significant limitations, failing to identify viable pathways for significant number of the target molecules while generating a substantial proportion of unreliable routes classified as *Nonsense* and *Rather Not*. IBM RXN shows improved performance by increasing the number of reliable pathways and reducing hallucinated predictions, yet fails to produce valid synthetic routes for a considerable fraction of target compounds. RetroChimera outputs pathways for the vast majority of targets. Although it produces a large percentage of Safe Bet pathways, it also generates a substantial number of Nonsense pathways, showing that ensembling current generators alone is not sufficient to mitigate errors for challenging targets.

RetroAligned without scoring significantly improves number of pathways found, providing solutions for all targets. However, confidence in its results is undermined by the significant presence of unreliable *Nonsenense* and *Rather Not* paths. Introducing individual scorers increases the fraction of targets for which no paths are found, a trade-off that can be desirable for the trustworthiness of the system — rejecting some targets is preferable to mixing reliable and unreliable pathways, as long as the remaining routes are correct. While RGP and RRS scorers reduce number of unreliable paths only modestly, our RP scorer demonstrates its value as a primary filter by eliminating all

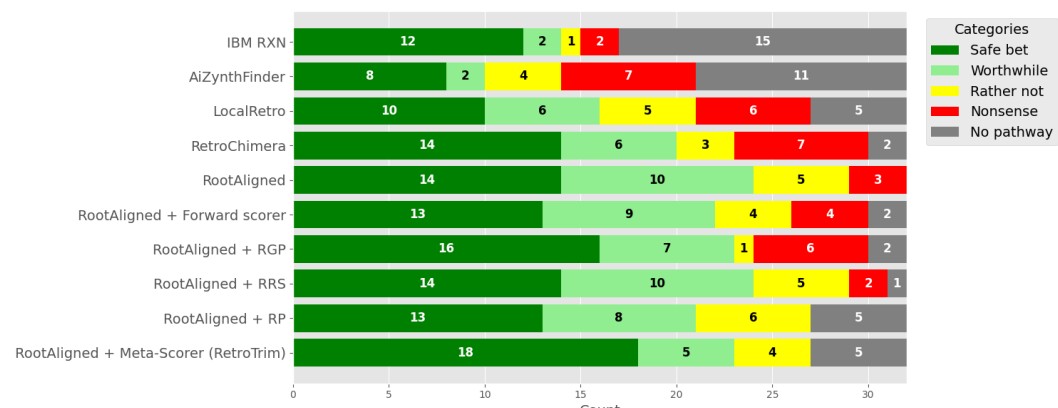

Figure 3: Comparison of our retrosynthesis generator (RootAligned) with different scorers against IBM RXN, AiZynthFinder, LocalRetro, and RetroChimera. Among AiZynthFinder, IBM RXN, LocalRetro and RetroChimera, RetroChimera performs significantly better than others, but it still fails on >25% targets, with a significant number of hallucinations. RootAligned without any reaction scorer finds pathways for all targets but includes unreliable routes. Introduction of individual scorers trades coverage for reliability, with RP eliminating all *Nonsense* pathways. RetroTrim, backed by the MetaScorer produces the most trustworthy results.

*Nonsense* reactions, though this comes at the cost of fewer *Safe Bet* and *Worthwhile* pathways. Finally, RetroTrim, utilizing the MetaScorer, delivers substantial improvements in reliability: significantly increasing *Safe Bet* paths, maintaining zero *Nonsense* results, and reducing *Rather Not* pathways. In our results, individual scorers similar to those commonly appearing in the literature (such as RGP - feasibility classifier and forward scorers), weren't sufficient to achieve correct pathways. RetroTrim provides the largest number of problematic routes while eliminating the most serious errors.

## 6.2 REACTION-LEVEL PLAUSIBILITY EVALUATION

Our results show that the MetaScorer outperforms individual scorers in both precision and recall, demonstrating effective integration of complementary signals. Figure 4 presents the ROC and precision-recall curves, with the MetaScorer achieving consistently higher area under the curve (AUC) values across both metrics. Similar curves broken down by reaction failure category can be found in Appendix E.

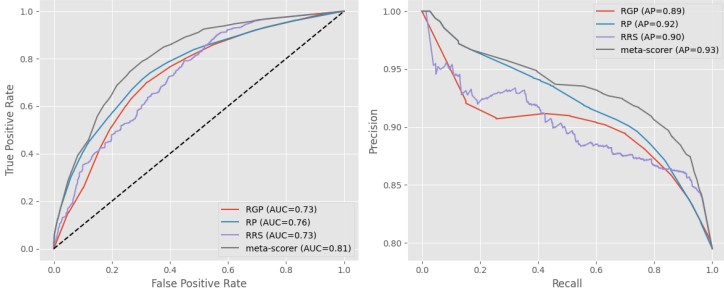

Figure 4: ROC (on the left) and precision-recall (on the right) curves comparing the performance of individual scorers versus the MetaScorer. The MetaScorer achieve higher AUC values for both ROC and PR curves, indicating better discrimination between plausible and implausible reactions. Among the individual scorers, RP shows the best performance.

Figure 5 shows ROC-AUC values for each scorer broken down into different failure categories, illustrating that individual scorers demonstrate proficiency in filtering out reactions deemed implausible under different evaluation criteria. RGP achieves the best performance on *Selectivity* and *Reactivity* errors. RRS is most capable of detecting fundamental structural issues such as *Reactant mismatches* and *Magic*, in addition to *One pot* errors. RP shows a balanced profile, which explains its overall

superior performance compared to RGP and RRS in Figure 4. By leveraging the unique strengths of each individual scorer, the MetaScorer maintains robust predictive performance across all failure categories.

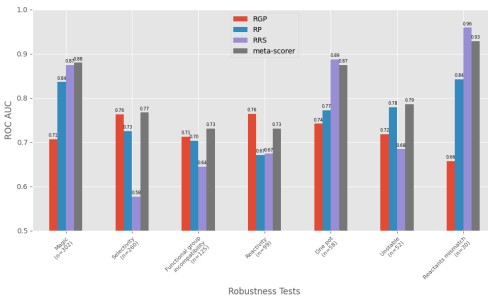

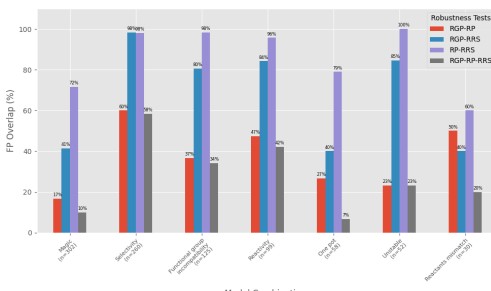

Figure 5: ROC-AUC performance of individual scorers across different failure categories, with sample sizes indicated for each category.

Figure 6: Overlap between individual pairs of scorers and triple of all scorers across different failure categories, with sample sizes indicated for each category.

We also analyze the overlap between false positives that each scorer fails to filter, as shown in Figure 6. The results show distinct complementarity: while RRS and RP exhibit high overlap in most categories, it is notably reduced for *One-pot*, *Magic*, and *Reactant mismatch* failure modes — the categories where Figure 5 demonstrates RRS's superior performance. RGP and RP show consistently low overlap across all failure categories, indicating that these scorers capture different aspects of reaction implausibility. Importantly, when considering all three scorers jointly, the overlap drops to very low levels across all categories, providing strong evidence that each scorer contributes unique discriminative value essential for building a robust MetaScorer.

## 7 CONCLUSIONS

In this work, we introduced RetroTrim, a retrosynthesis system designed to avoid hallucinations in the synthesis tree through a combination of three complementary reaction scoring strategies. We demonstrated its success on thirty-two unpublished drug-like targets, where no generated paths contained hallucinated reactions. Among the available methods we compared RetroTrim with, our method was the only one to compltetely avoid hallucinations, while at the same time finding more paths without issues than other methods. To understand the strengths and complementarity of each scoring strategy, we compared their performance across different classes of possible issues. We found evidence of synergy between the scorers, both at the level of filtering individual reaction, and in terms of the retrosynthetic paths resulting from their use.

In evaluating retrosynthesis systems and scorers, we made use of a novel labeling protocol where we leveraged chemists' expertise to produce fine-grained labels for generated reactions. To our knowledge, this is the first instance of such a granular analysis of retrosynthetis systems' output, where automated metrics and ad-hoc manual inspection were the norm. In order to facilitate further development in the field, we release the thirty-two targets used for path generation. While our evaluation process is generally applicable to retrosynthesis, RetroTrim was trained on data that biases it towards the medicinal chemistry context. We also note that by focusing on the top-1 performance of retrosynthesis systems, we leave open for further work the analysis of how different plausibility filtering methods impact the diversity of resulting paths. Nonetheless, we hope that the insights and methodologies presented in this work lead to more reliable retrosynthesis in general.

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

# A EXAMPLES OF REACTION PATHWAYS

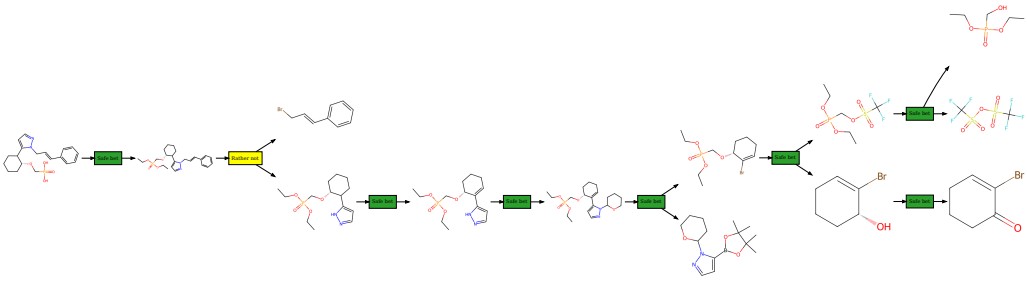

Figure 7: Example of a pathway with Safe Bet and Worthwhile reactions

Figure 8: Example of a pathway with a Rather Not reaction

Figure 9: Example of a pathway with a Nonsense reaction

## B GAT

The GAT model processes reaction graphs where individual atoms and bonds are featurized with chemically-meaningful characteristics, outputting a scalar plausibility score for each reaction. The attention mechanism is modified to ensure that attention weights between non-connected nodes approach zero, preserving the chemical connectivity structure. The key difference to the original GAT is the support of global information exchange across the entire molecular graph, ensured by an artificial supernode that connects to all other nodes in the graph.

## C REACTION EVALUATION PROTOCOL

Each candidate reaction is assessed sequentially along the following dimensions. Unless specified otherwise, in each of them the reaction is scored on a four-level confidence scale: *Nonsense*, *Rather Not*, *Worthwhile*, and *Safe Bet*, indicating the plausibility of the reaction. Every reaction that is not a safe bet is assigned an additional label explaining the reason for its incorrectness.

1. **Reactant-Product Consistency:** Structural alignment between reactants and product is verified. Reactions in which the product contains a large fragment that is neither present in the substrate nor originates from a commonly used reagent, or in which no clear relationship between the atoms of the product and the substrates can be established, are marked as *Nonsense*, with the reason for incorrectness labeled as *Reactants mismatch*.

2. **Stability:** Reactions producing products or including substrates that are not isolable under the typically achievable conditions are marked as *Nonsense*, with the reason for inplausibility labeled as *Unstable*.

3. **Mechanistic Plausibility:** Reactions lacking a plausible mechanism are classified as *Nonsense* or *Rather Not* due to *Magic*, covering transformations requiring unknown or highly implausible reactivity. Transformations that would require more than two non-trivial steps are also placed in this category.

4. **Multistep Feasibility and One-Pot Potential:** Reactions not achievable in a single step are assessed for decomposability into two coherent steps. If they pass this test, feasibility in a one-pot setting is scored on a four-level scale and failing reactions are marked as *One pot*.

5. **Reactivity of Substrates:** Feasibility of the reaction, given the reactivity of the substrates (e.g., electron deficiency), is verified. Reactions that cannot be reasonably expected to occur are marked as implausible, with the reason for incorrectness labeled as *Reactivity*.

6. **Functional Group Compatibility:** Molecules are screened for other functional groups that can undergo a reaction. If other groups are more probable to react first, the reaction is marked with problem *Functional group incompatibility*.

7. **Selectivity:** Selectivity of the reaction is verified, including competition between functional groups of the same type, regioisomeric outcomes (e.g., in electrophilic aromatic substitution), or other cases where multiple plausible products can arise. Reactions that fail this evaluation are marked as *Selectivity*.

## C.1   TYPES OF ERRORS IN PATHWAYS GENERATED BY EVALUATED SYSTEMS

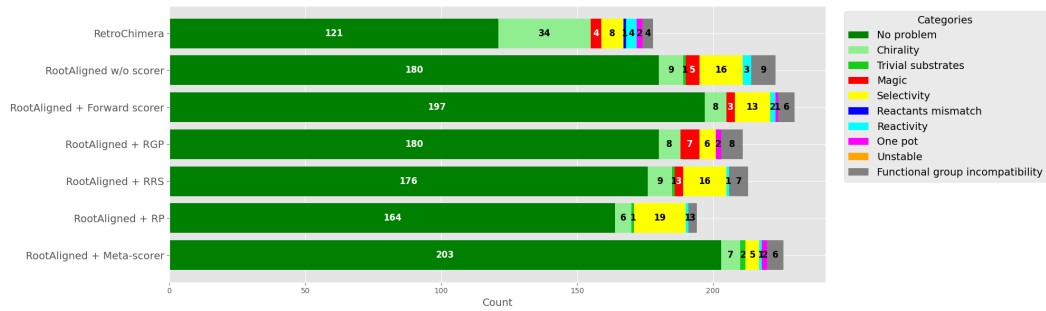

Figure 10: Detailed information on the types of errors for each reaction in the pathways from Fig. 3. Note that only reactions from the found pathways are presented here.

## C.2   IMPLAUSIBILITY ANNOTATION EXAMPLES

### C.2.1   REACTANTS MISMATCH

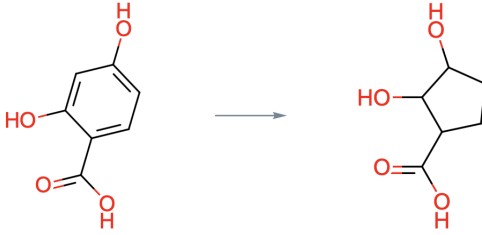

Figure 11: Nonsense: No clear relationship between atoms in the product and the substrate can be confidently proposed

Figure 12: Nonsense: The pyridyl fragment require an additional substrate, that is missing

## C.2.2 UNSTABLE

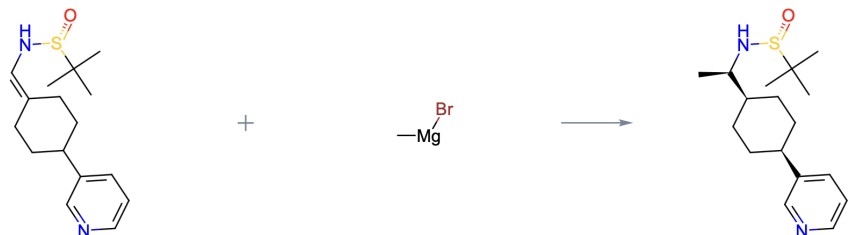

Figure 13: Nonsense: The carbon atom with amine and chlorine is not something seen in literature

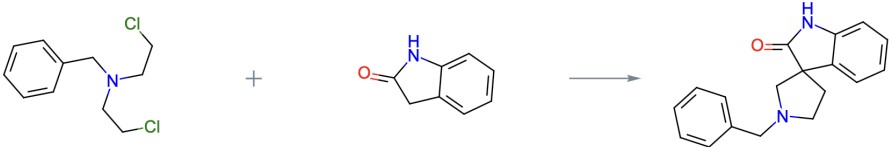

Figure 14: Nonsense: The second substrate would tautomerize to phenol instantly

Figure 15: Nonsense: The substrate is unstable, it would tautomerize to imine

## C.2.3 MAGIC

Figure 16: Nonsense: Changing length of the alkyl chain, no known precedent of such variant of carbon alkylation

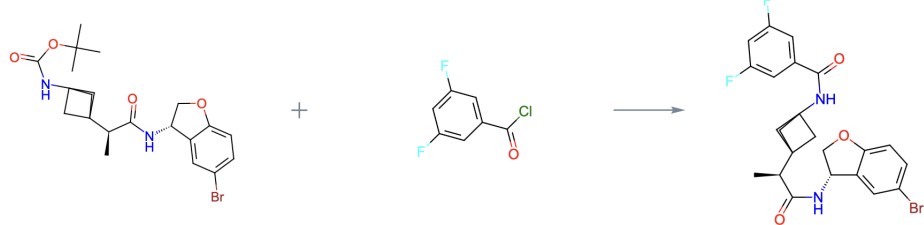

Figure 17: Nonsense: An alkyl chain acting as a leaving group and bond formation by an unactviated amine carbon. No such reactivity ever demonstrated in literature

### C.2.4 ONE POT

Figure 18: Rather not: 2 steps required – Boc deprotection and acylation

Figure 19: Rather not: 2 steps required - Cbz deprotection and Boc protection

### C.2.5 REACTIVITY

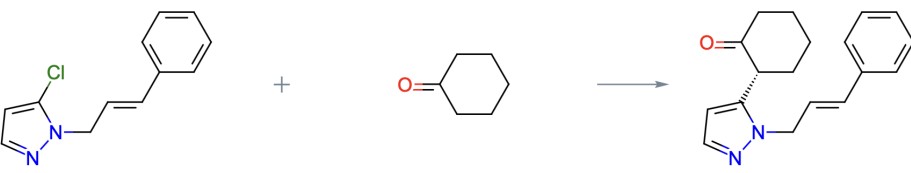

Figure 20: Rather not: Most of the references for this reaction are around electron-deficient heterocycles, only one example with pyrazole in literature

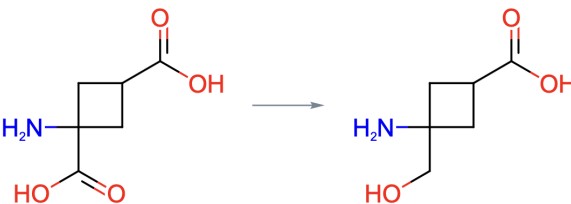

Figure 21: Rather not: High likelihood of steric hindrance

### C.2.6 FUNCTIONAL GROUP INCOMPATIBILITY

Figure 22: Rather not: No literature references where a bromine is located in alpha to the ester position. The alkyl bromine would most likely react more readily than the ester.

Figure 23: Nonsense: No conditions allow to cleave a methyl ether in a way that wouldn't affect the sulfonyl chloride

### C.2.7 SELECTIVITY

Figure 24: Rather not: There is a considerable risk that achieving the disubstituted product at a satisfactory yield would be very difficult (especially accounting for the presence of amine in the structure).

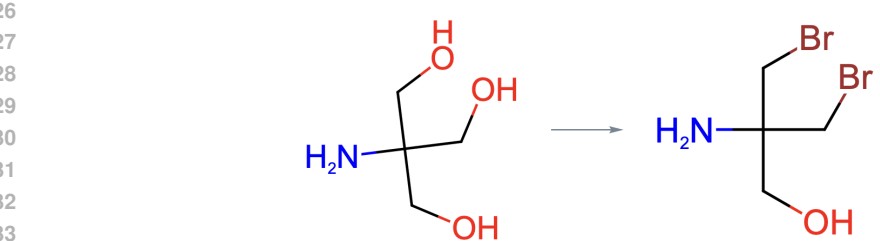

Figure 25: Rather not: There are 3 equivalent hydroxyl groups, so in bromination we expect triple substitution rather than this scenario

# D  RETROSYNTHESIS TARGETS

## D.1  SMILES

Clc1ccc(-c2c(N(CC)CC)c(c(nc2C)C)CC(=O)NCC)cc1

O(c1cc(c([N+](=O)[O-])cc1)COC1CN(C(=O)[C@@H]2C[C@]3(NC(OC3)=O)C2)C1)C1CCCC1

FC1(F)C(N2N=CC(=C2C)c2cc(ccc2)C#Cc2c(OC)cc(nc2)C(=O)O)C1

O(C(C)(C)C)[C@H](C(=O)Nc1nc2[C@](O)(CCc2cc1)CC)c1c(nc(cc1)C)C

FC1(F)Oc2c(O1)cc(nc2)C(=O)NC1=NN2C(C(=O)N[C@@H]3[C@H]2CCC3)=C1

Clc1c(N2CCC(F)(F)CC2)c(Cl)cc(NC(=O)CC[C@]2(NC(=O)NC2=O)C2CC2)c1

S(=O)(=O)(Nc1nc2N(N(C(=O)c2cn1)CC=C)C)c1ccc([C@@H](C2=Nc3c(N2)cccc3)CCO)cc1

Fc1cc(F)cc(N2[C@H](CN(CC(=O)Nc3ncnc4N(C(C)C)C=C(F)c34)CC2)C)c1

Fc1c(nc2c(c(F)ccc2)c1)Nc1cc2C(OC(=O)c2cc1)(C)C

O1C(=NN=C1)c1c(ncnc1)NC1C[C@H](O)[C@@H](O)C1

Fc1cc2c(OB(O)[C@@H](NC(=O)C3CC3)C2)cc1

S(C=1NN=NC1C(=O)NCCOCCNC(=O)C=1N=C(SC1)N1N=CC(=C1)C)c1ccccc1

O1C(Oc2c1c(ccc2C)C)([C@@H]1CC[C@@H](NC(=O)c2ncc(cc2)C#N)CC1)C

S(C1=C(C(=O)NC(=C1)C)CN(c1c2c(nccc2)c(cc1)C#N)C)C

O(CC(=O)NC1CC2N(C(C1)CC2)C)CCN1c2c3N(C(=O)C1=O)CCCc3ccc2

FC(F)(F)c1cc(C2=CN(C(=O)C(NC(=O)C3=NN(c4c3cccc4)C)=C2)C)ccc1

Fc1cc(F)cc(C(=O)NC23CC([C@@H](C(=O)N[C@H]4c5c(OC4)ccc(-c4c(OC)ccc(c4)C)c5)C)(C2)C3)c1

O1c2c(cc(C3=CN4N=C(N=C4N=C3)c3cnc(C(=O)C)cc3)cc2)CCC1

S1C(N(C(=O)C2C(OCC)C=CCC2)C)=C(C2=C1CC1(N(C2)CC2CC2)CCCC1)C#N

S(=O)(=O)(N[C@@H]([C@@H]1CC[C@H](c2cnccc2)CC1)C)c1cc(F)cc(-c2ncccc2)c1

FC(F)(F)[C@@H](N1CCC2(C(=O)N(Cc3c4OC=C(c4cc(OC(C)C)c3)C)CC2)CC1)CC1[C@@H](O)[C@@H](O)CC1

FC(F)(F)[C@@H]([C@H](C(=O)N[C@@H]([C@@](O)(N)CC)C)c1cc(OC)cc(OC)c1)C

FC(F)(F)c1ncc(-c2ncc(C(F)(F)F)c(c2)CNC(c2cc(C3=NOC(=C3CO)CC)ccc2)C2CC2)cn1

O1c2c(nc(N3C(=CC=C3C)C)nc2CCC1)NC1CCC(CO)CC1

O(c1ccc([N+](=O)[O-])cc1)CC[C@@](N)(CCN(C(=O)c1c2c(C(=O)c3c(C2=O)cccc3)ccc1)C)C

S(=O)(c1ccccc1)CCNC(=O)CN(c1ncnc([C@@H]2C[C@@H](O)C2)c1)C

Fc1c(C=2OC(=NN2)C=2Oc3c(cc4NC(Oc4c3)=O)C2)cc(F)cc1

O=C(N1C2C(Nc3ncc(-c4cnccc4)cn3)CC1CC2)C1C(O)C(O)CC1

S1[C@]2(C(=O)N3CC4[C@@H](NC5=NN(C=N5)CC(F)(F)F)[C@H](C3)CC4)[C@H]([C@](N=C1N)(c1ccccc1)C)C2

P(=O)(O)(O)CO[C@H]1C(C=2N(N=CC2)C/C=C/c2ccccc2)CCCC1

FC(F)(F)C(Nc1cncc(C(CO)C)c1)c1c(F)cc(OC2CN(C2)CCCF)cc1

O=C(N1CC(N2C(=O)CNC(C2)C)C1)N[C@H]1C(=O)NC[C@@H]1c1ccc(N2C[C@@H](O)CC2)cc1

## D.2 VISUALIZATION

Figure 26: 32 molecules that have been used as targets for retrosynthesis.

## E   ROC AND PRECISION-RECALL CURVES BY FAILURE CATEGORY

### E.1   MAGIC

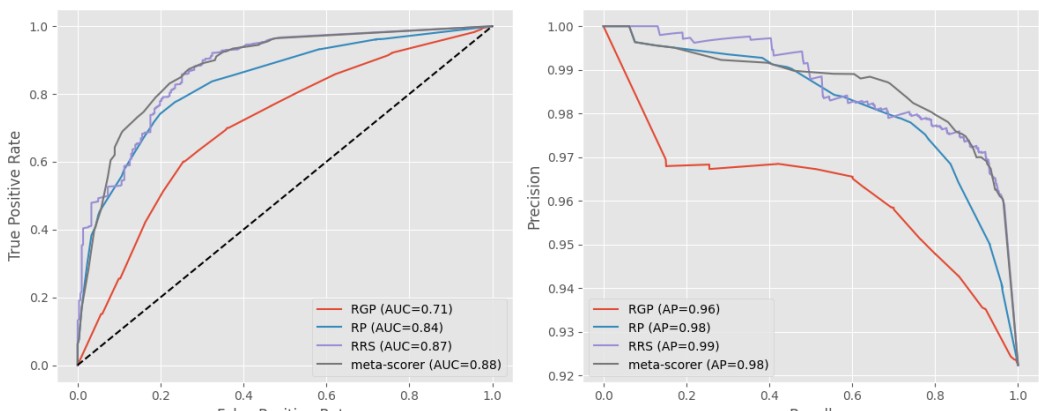

Figure 27: ROC (on the left) and precision-recall (on the right) curves comparing the performance of individual scorers versus the MetaScorer on *Magic* and *No Problem* reactions.

### E.2   SELECTIVITY

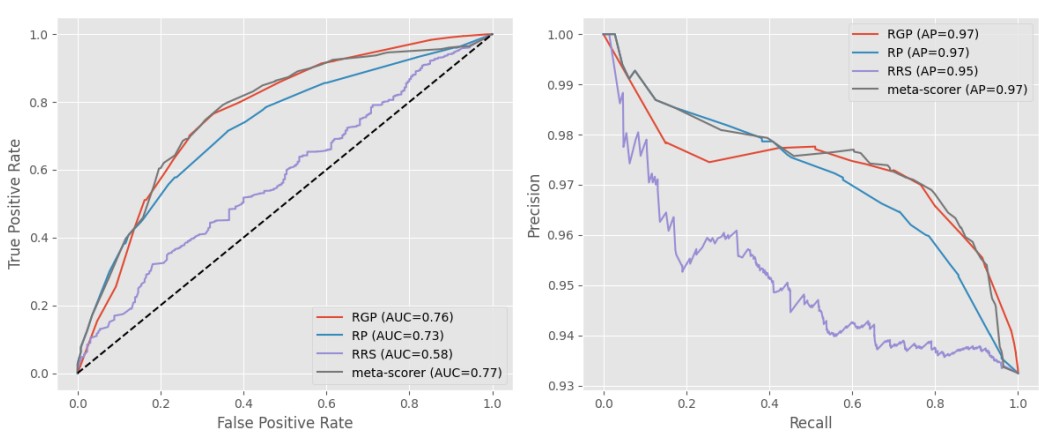

Figure 28: ROC (on the left) and precision-recall (on the right) curves comparing the performance of individual scorers versus the MetaScorer on *Selectivity* and *No Problem* reactions.

## E.3 FUNCTIONAL GROUP INCOMPATIBILITY

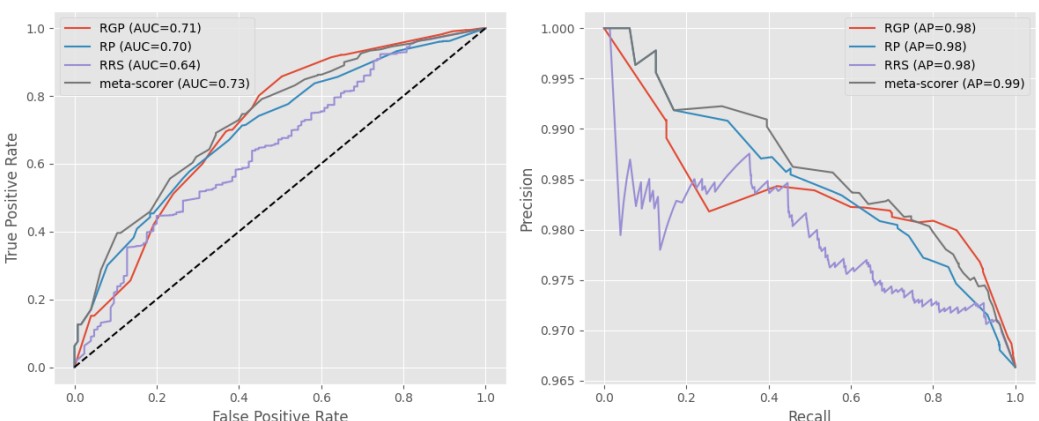

Figure 29: ROC (on the left) and precision-recall (on the right) curves comparing the performance of individual scorers versus the MetaScorer on *Functional group incompatibility* and *No Problem* reactions.

## E.4 REACTIVITY

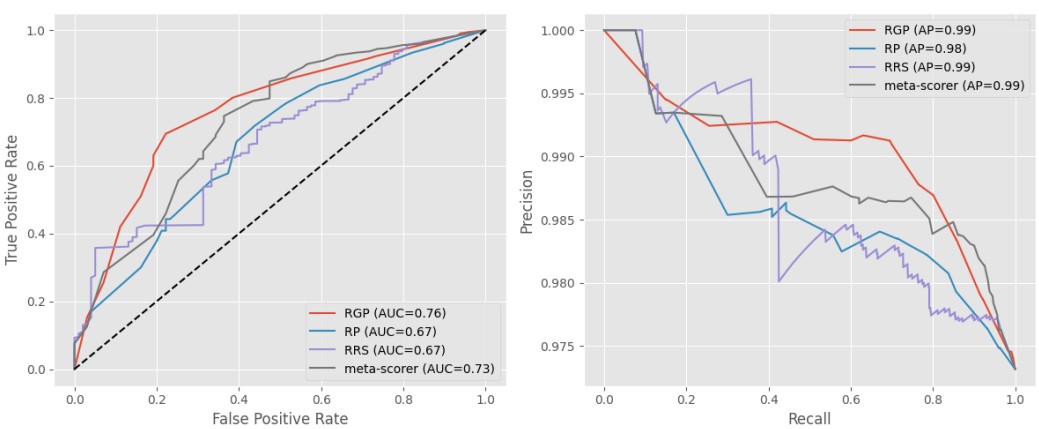

Figure 30: ROC (on the left) and precision-recall (on the right) curves comparing the performance of individual scorers versus the MetaScorer on *Reactivity* and *No Problem* reactions.

## E.5 ONE POT

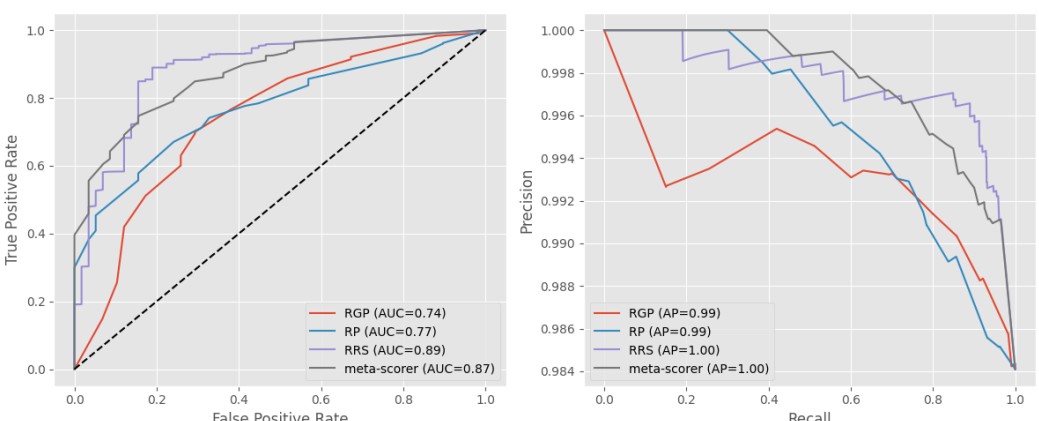

Figure 31: ROC (on the left) and precision-recall (on the right) curves comparing the performance of individual scorers versus the MetaScorer on *One pot* and *No Problem* reactions.

## E.6 UNSTABLE

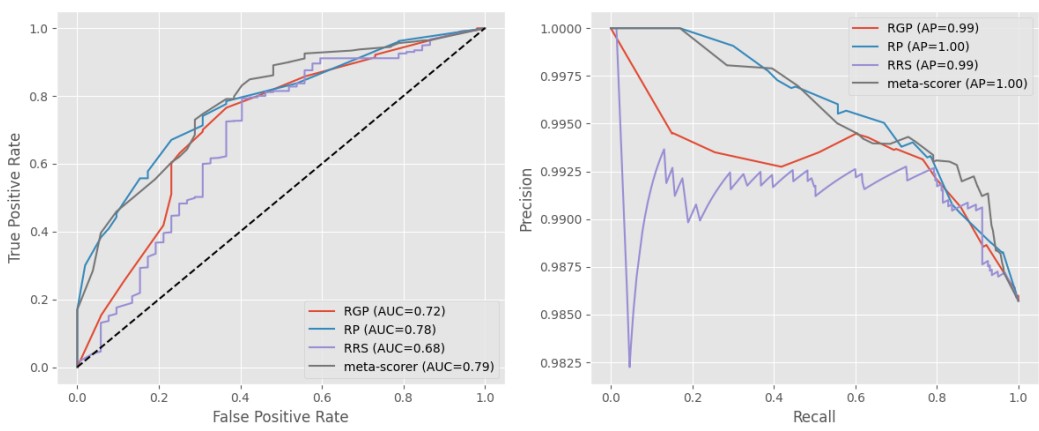

Figure 32: ROC (on the left) and precision-recall (on the right) curves comparing the performance of individual scorers versus the MetaScorer on *Unstable* and *No Problem* reactions.

## E.7 REACTANTS MISMATCH

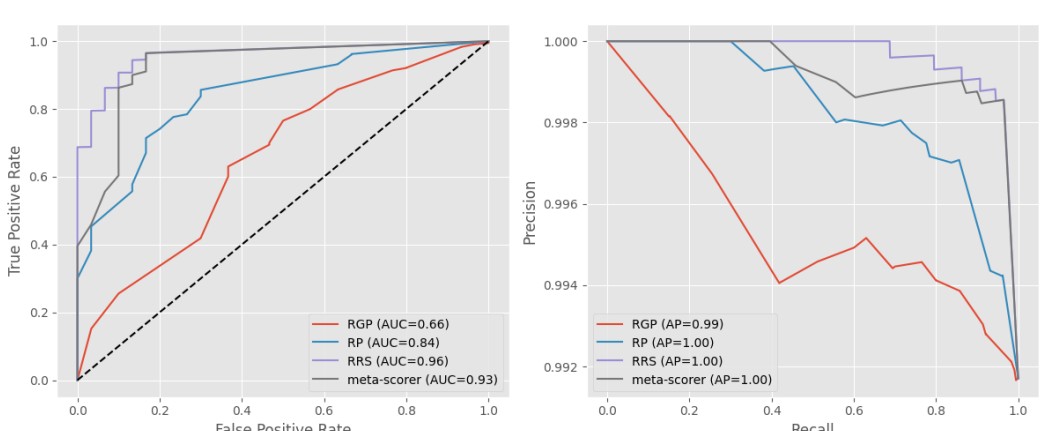

Figure 33: ROC (on the left) and precision-recall (on the right) curves comparing the performance of individual scorers versus the MetaScorer on *Reactants mismatch* and *No Problem* reactions.

# F FALSE POSITIVES COUNTS BY FAILURE CATEGORY

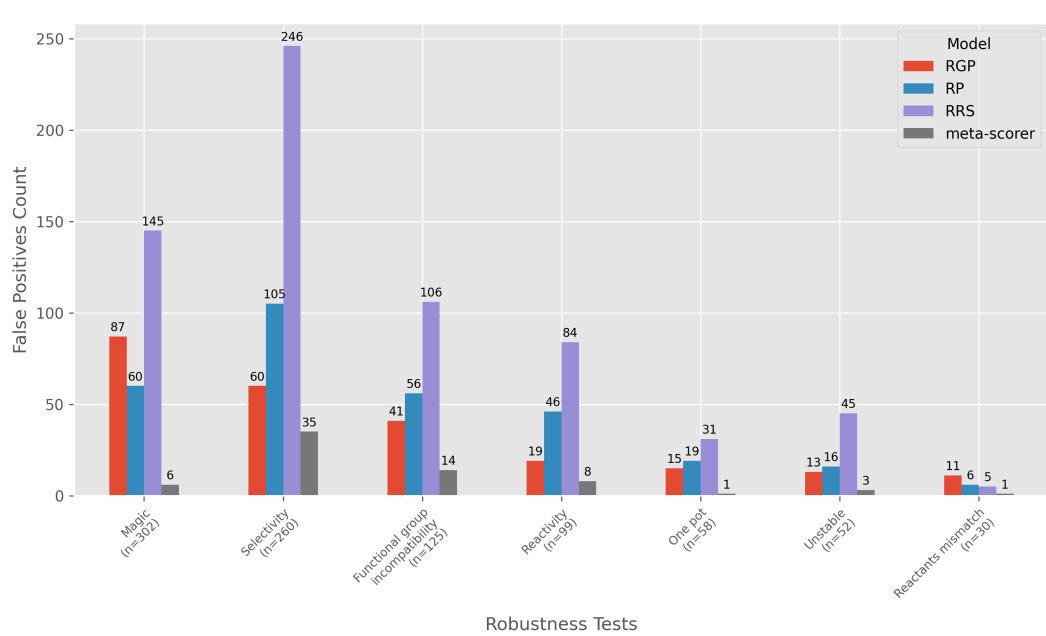

Figure 34: Counts of false positives produced by individual scorers versus the MetaScorer across different failure categories, with sample sizes indicated for each category.

# G TRUE NEGATIVES COUNTS BY FAILURE CATEGORY

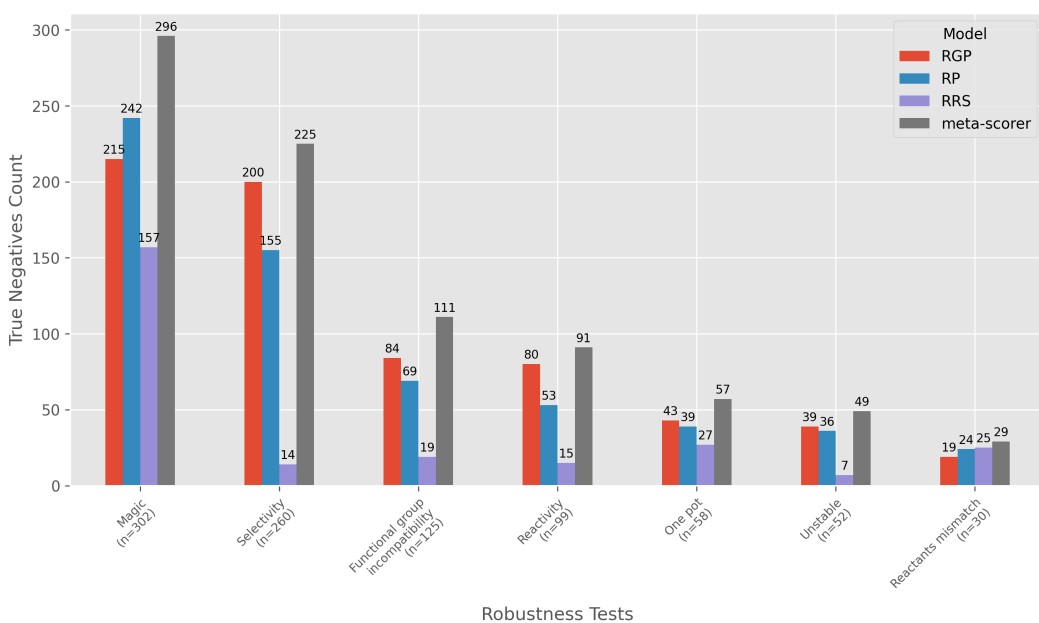

Figure 35: Counts of true negatives produced by individual scorers versus the MetaScorer across different failure categories, with sample sizes indicated for each category.

# H LARGE LANGUAGE MODEL USAGE

We used large language models solely to polish the writing by correcting grammar and spelling errors. No part of the technical content, methodology, or results was generated or influenced by these models.

