# OpenReview forum: "Trustworthy Retrosynthesis: Eliminating Hallucinations with a Diverse Ensemble of Reaction Scorers"
_ICLR.cc/2026/Conference — Submitted to ICLR 2026_

### Official Review · Reviewer_38oB · 2025-10-27

**Soundness:** 2
**Presentation:** 2
**Contribution:** 2
**Rating:** 2
**Confidence:** 3

**Summary:**

The authors propose a post-processing approach for retrosynthesis model outputs that employs three filters, which are then integrated into a meta-filter to refine and eliminate implausible reactions.

**Strengths:**

- The method is well-motivated and logically structured.
- The approach effectively incorporates substantial human chemical expertise.

**Weaknesses:**

### **Major Comments**
1. **Scope of Hallucination Types**
   The abstract claims that the method can “capture different classes of hallucinations.” However, in the main paper, only one type—*Nonsense* hallucinations—is addressed. The authors should clarify whether other classes were considered or provide justification for focusing only on one type.

2. **Claim of Avoiding All Hallucinations**
   The claim that RetroTrim “avoids all hallucinated reactions” appears overstated. Figure 4 suggests that the outcome heavily depends on the threshold and likely other hyperparameters. The authors should discuss how recall and precision are balanced to produce the results in Table 3—for example, how the threshold value is selected, and what trade-offs exist between filtering false positives and removing valid reactions.

3. **Design and Normalization of RP Scores**
   The rationale behind the design choices for the reaction plausibility (RP) metrics requires clarification. Why is \( S_{RP} \) normalized by \(\sqrt{T}\), while \( S_{RC}\) is normalized by \(T\)? How were these normalization schemes determined? Furthermore, practical guidance on how to choose the parameters \(\alpha\), \(\beta\), and \(\gamma\) would improve the applicability of the method.

4. **Dataset Reproducibility**
   Although the Pistachio dataset offers advantages over USPTO, it is proprietary and not publicly accessible, which limits reproducibility. The authors are encouraged to also evaluate their method on the USPTO dataset to enhance comparison and ensure open benchmarking.

5. **Missing Reference in Figure 3**
   The caption of Figure 3 mentions “Our generator w/o scorer,” but this variant does not appear in the figure. Please verify and correct the figure or the caption.

---

### **Minor Comments**
- Some reference formatting issues exist (e.g., missing brackets in lines 73, 318, 319).
- Text in Figures 5 and 6 is too small to read and should be enlarged for clarity.

**Questions:**

see weaknesses.

---

> ### Author Response · Authors · 2025-11-21
> **Thank you for your review**
>
> We thank the Reviewer for the helpful comments.
>
> We wanted to start by emphasizing hallucinated reactions are, in our experience, the main practical obstacle to deploying neural retrosynthesis systems. In a multi-step route, a single chemically nonsensical step (for example, a transformation that creates or breaks several bonds without any plausible mechanism or required reagents) effectively invalidates the entire pathway, even if all other steps look reasonable. Standard metrics such as top-k match to a reference route or overall search success rate tend to hide this issue, because they do not require that every proposed step be something a chemist would be willing to try in the lab. Our work is explicitly centered on detecting and removing these nonsensical steps, which is why we invest significant effort into defining and annotating hallucinations at the reaction level.
>
> For example, in Figure 1 of the paper **the SSR model proposes an ortho-amino benzoate directly converting into a triazole ring.** A PhD-level chemist can see that this does not belong to any known reaction class and that no reasonable set of conditions or reagents is available in the literature; **executing it would require developing a completely new methodology, i.e., a multi-month research program rather than a practical synthetic step**. Yet such reactions appear in the top-1 routes of existing systems. This is analogous to LLMs: high benchmark accuracy does not prevent hallucinations on out-of-distribution, real-world queries.
>
> **Response to Weaknesses**
>
> **W1: Scope of hallucination**
>
> The Nonsense category encompasses seven types of errors; if any of these is serious, the reaction is labeled with this aggregate Nonsense tag. Please see the “Reaction Evaluation Protocol” in the Appendix. In other words, we are not focusing on a single narrow error mode: any step that a PhD-level chemist judges as effectively infeasible in a single operation (e.g., requiring multi-step methodology development, impossible selectivity, or incompatible functional groups) is treated as a hallucination via this Nonsense label. We would be happy to bring the key parts of this protocol into the main text to make this clearer.
>
> **W2: Claim of avoiding all hallucinations**
>
> Thank you for your comment. We agree this is a precision–recall tradeoff. In the camera-ready version we would be happy to include an explicit analysis of this tradeoff (e.g., by varying the threshold), but this is too costly to run during the rebuttal phase due to the need for additional human annotation. However, the results as currently reported already show a large and statistically significant difference between methods in terms of hallucination rate, which is the basis for our claim.
>
> **W3: Design and normalization of RP scores**
>
> Reaction Prior was designed from a chemistry-first perspective, and we are sorry that the paper did not do a better job communicating this.
>
> Reaction Prior Score (RS): LLMs are known to assign higher probability to sequences close to the training set. Therefore, this component acts as a measure of similarity to the training set, mimicking how a chemist would look for precedent for the overall transformation.
>
> Reaction Center Score: Same as RS but focused on the reaction center. Chemists tend to pay special attention to the reactive core, so we explicitly weight this region.
>
> Regioselectivity Score: We compare the plausibility of different potential regioisomers. If there is precedent in the training set for a particular regioisomer, its RS will be higher.
>
> We hope this explanation makes the intuition clearer, and we will improve the paper accordingly. The exact normalization (e.g., division by √T vs. T) is treated as a hyperparameter and was chosen empirically based on predictive performance, which has good precedent in the LLM literature.
>
> **W4: Dataset reproducibility**
>
> We agree. We will be happy to add an evaluation on USPTO in the camera-ready version to improve reproducibility and open benchmarking. Again, running a full human-labeled USPTO evaluation during the rebuttal phase is too expensive due to annotation cost.
>
> **W5: Missing reference in Figure 3**
>
> We are sorry for this oversight. We will fix Figure 3 (and its caption, if needed) so that all variants mentioned in the text are consistently represented.

---

> > ### Comment · Reviewer_38oB · 2025-11-28
> >
> > Thank you for the response. I am happy to reconsider the score once the revised version is available.

---

> > > ### Author Response · Authors · 2025-12-03
> > > **Thank you**
> > >
> > > Thank you for being open to this.
> > >
> > > We have uploaded a new version addressing all of your comments.
> > >
> > > Most importantly, we wanted to draw your attention to the practical importance of hallucinations. The main contribution of the paper is dramatically reducing hallucination (the Nonsense category) that plagues generative models for text/vision (LLMs) and chemistry alike. We have updated Figure 2 and Introduction to better illustrate this point. We also include comparison to recent work such as RetroChimera.

---

### Official Review · Reviewer_qjSH · 2025-10-27

**Soundness:** 2
**Presentation:** 2
**Contribution:** 2
**Rating:** 2
**Confidence:** 5

**Summary:**

Single-step predictions in multi-step retrosynthesis often accumulate false positives because single-step models remain imperfect (typical top-1 ≈60%). To assess true multi-step accuracy, this paper presents RetroTrim, an evaluation protocol that scores each proposed reaction by combining machine-learning models with reaction databases. At its core is a Reaction Prior scorer, complemented by two additional scorers tailored to distinct error modes. These three signals are then aggregated into a Meta-Scorer that delivers a robust, final judgment of reaction correctness for each step—and, by extension, the overall route.

**Strengths:**

1. The paper tackles a real deficit in multi-step retrosynthesis evaluation. It highlights that single-step models are not oracles (typical top-1 ≈60%) and that search success rate is an unreliable metric. The work introduces a new evaluation metric specifically designed to address these shortcomings.

2. The study validates the proposed metric(s) with expert assessment, reducing reliance on potentially biased or error-prone model signals.

**Weaknesses:**

1. The conceptual rationale behind the proposed metrics (Reaction Prior Score etc.) is not fully articulated, making it hard to understand why they should correlate with true reaction correctness.

2. The sample set used for assessment is small, weakening the statistical confidence in the method’s reported accuracy and robustness. 4500 reactions and 32 samples are not enough.

3. The work does not compare against standard molecule synthesizability indicators (e.g., SA score), leaving open whether the metric adds value beyond established proxies.

**Questions:**

N/A

---

> ### Author Response · Authors · 2025-11-21
> **Thank you for your feedback**
>
> We would like to first clarify the problem we address. In multi-step retrosynthesis, even a single hallucinated step can invalidate an entire route. These are not just “top-1 ≈ 60%” errors on standard benchmarks, but chemically nonsensical transformations on realistic, drug-like targets. For example, in Figure 1 of the paper the SSR model proposes an ortho-amino benzoate directly converting into a triazole ring. A PhD-level chemist can see that this does not belong to any known reaction class and that no reasonable set of conditions or reagents is available in the literature; executing it would require developing a completely new methodology, i.e., a multi-month research program rather than a practical synthetic step. Yet such reactions appear in the top-1 routes of existing systems. This is analogous to LLMs: high benchmark accuracy does not prevent hallucinations on out-of-distribution, real-world queries.
>
> **Response to Weaknesses**
>
> **W1. Conceptual rationale behind Reaction Prior Score**
>
> Reaction Prior was designed from a chemistry-first perspective, and we agree the paper did not communicate this clearly enough.
>
> Reaction Prior Score (SRP): Autoregressive models (including LLMs) assign higher probability to sequences close to their training distribution. SRP is the normalized log-likelihood of the whole reaction, acting as a global measure of similarity to known chemistry and mimicking a chemist’s search for precedent.
>
> Reaction Center Score (SRC): Same idea, but restricted to the reaction center tokens. Chemists pay particular attention to the atoms that actually change; SRC reflects the plausibility of those local changes.
>
> Regioselectivity Score (SRegio): We compare the probability of the desired regioisomer to the summed probability of alternative regioisomers. If there is precedent for a specific regioisomer in the training set, its SRP will be higher, and SRegio captures that preference.
>
> We hope this makes the design more intuitive; we will revise Section 3 to present this rationale more clearly.
>
> **W2. The sample set size**
>
> We understand the concern. However, to our knowledge this is the first chemically grounded, step-level evaluation of multi-step retrosynthesis at this scale. Our 32 targets expand into over 200 individual reaction steps. Each step was carefully reviewed by an expert chemist, including a comparison against literature precedents. Doing this thoroughly is extremely time-consuming, and scaling it up without losing annotation quality is very challenging.
>
> **W3. Lack of comparison to SA score**
>
> We respectfully disagree that SA score is an appropriate comparison here. SA scores aim to quantify how “easy” a molecule is to make in general, not whether a specific proposed reaction step is chemically correct. Our goal is to evaluate and improve the correctness of full, model-generated synthetic pathways that can plausibly work in the lab. Existing retrosynthesis papers at top venues also typically do not compare against SA scores for this reason. We will clarify this distinction in the revised version.

---

> > ### Comment · Reviewer_qjSH · 2025-11-22
> >
> > Thanks for the detailed response.
> >
> > I do not think using the Reaction Prior Score, Reaction Center Score, or Regioselectivity Score to assess the feasibility of a retrosynthesis prediction is reasonable. Since I think they are not directly related to the feasibility of a retrosynthesis prediction.  At present, aside from exact matching, we lack rigorous evaluation criteria, and in my view, these three scores are largely unrelated to whether a proposed synthetic route is actually feasible.
> >
> > More broadly, evaluating the feasibility of retrosynthesis predictions is a fundamentally difficult and still open problem. In my four years working in this area, I have not seen a truly satisfactory metric. That said, I believe round-trip evaluation using a forward reaction model is more promising, since forward models are generally quite accurate. The main limitation, however, is the absence of a sufficiently large and high-quality dataset to support a very strong forward reaction model.
> >
> > Overall, this paper appears to be a good attempt in that direction, but I remain unconvinced that the proposed approach will be effective in real-world production settings.

---

> ### Author Response · Authors · 2025-11-23
> **Thank you**
>
> We appreciate the quick answer.
>
> Respectfully, but on what data is the Reviewer's opinion that Forward model is better than Reaction Prior based on? Please note that results in the paper are consistent with our claim:
>
> **Our main benchmark in Figure 3 shows that RootAligned + Forward produces 13% hallucinated pathways, compared to 0% of RootAligned + MetaScorer**
>
> Please further note we use the ground truth -- evaluation of PhD level chemists -- compared to computational (and hence necessarily heuristics) methods such as the round trip accuracy. From this perspective we are also confused by the Reviewer comment that we focus on developing a metric (which we are not, we use ground truth as metric).
>
> Finally, please note that Reaction Prior Score can be seen as a direct extension of Forward Model, given that it predicts the likelihood of the product given the substrates.
>
> Is there any critical experiment (please note that we are limited by expensive human evaluations) that is missing from the perspective of the Reviewer?

---

> > ### Comment · Reviewer_qjSH · 2025-11-24
> >
> > Could you provide or clarify the implementation details of the Forward Scorer?
> >
> > Is “evaluation of PhD-level chemists” as a primary criterion reasonable? For determining whether a reaction exists, we instead directly check its presence in some literature libraries. If you consider “evaluation of PhD-level chemists” to be a reasonable standard, then this work may be more appropriate for a chemistry journal, where such expert judgment is more common, rather than for a machine learning conference. In the context of a machine learning venue, the most reliable way to validate reaction correctness as ground truth is through exact matching.

---

> ### Author Response · Authors · 2025-11-26
> **Thank you**
>
> We again thank for the prompt response of the Reviewer.
>
> **Scorer implementation details**
>
> We use all scorers, including the forward scorer, in the same way: during a search we reject reactions that have an assigned score below a tuned threshold. The model is the same encoder-decoder Transformer as Reaction Prior, but has changed its objective to predict product given the substrates vs both substrates & product (as in the case of Reaction Prior).
>
> **Human-based evaluation**
>
> The dogma' to focus only on topK matching reactions restricts the field dramatically to the biased set of reactions where there exists an experimentally verified synthetic protocol to make the product. As a practitioner in the field, you are aware that in practice, a chemist might input a hard-to-make compound for which some steps do not have a good retrosynthetic cut. These are out of distribution in machine learning terms.
>
> We also dramatically bias the evaluation to the space of reported reactions. As a long-term practitioner in the field, haven't you noticed the dramatic discrepancy between top-1 accuracy of these models vs the practical accuracy and utility of the pathways?
>
> Please note that in Figure 1 of the paper, the SSR model proposes an ortho-amino benzoate that directly converts into a triazole ring. A PhD-level chemist can see that this does not belong to any known reaction class and that no reasonable set of conditions or reagents. A model achieving stellar top-1 accuracy proposed this transformation.
>
> (We would be happy to add this discussion to the paper.)
>
> Again, please let us know if there is a critical experiment that is missing. Respectfully, otherwise, we are just repeating dogma in the field and base our evaluation of this paper on opinions vs facts.
>
> ' Finally, **please note that the most important papers in the field have used human evaluation such as Segler et al Nature paper.**

---

> > ### Comment · Reviewer_qjSH · 2025-11-27
> >
> > Using the reaction model’s confidence as the Reaction Prior Score is not well-justified. A high score only indicates that the prediction is consistent with the model’s learned distribution; it does not demonstrate that the prediction itself is chemically correct.
> >
> > Human expert evaluation is inherently biased. In principle, the most reliable approach would be experimental validation, but this is prohibitively expensive at scale. As a more practical and objective alternative, I believe checking whether the predicted reactions exist in established chemical reaction databases is a more reasonable way to define ground truth. Even for work published in high-profile venues such as Nature, I personally think that strict and transparent evaluation standards should be upheld, regardless of the conference or journal.
> >
> > For the forward scorer, I believe a better strategy is to directly test whether the reaction model can reconstruct the product, rather than relying on a confidence threshold.
> >
> > I am willing to increase my overall score to 4.0. This paper is a valuable attempt, but in my view it still does not meet the standard required for publication at this venue.

---

> ### Author Response · Authors · 2025-11-27
> **Thank you**
>
> We thank the Reviewer again for engaging in the discussion and raising the score. However, we are afraid that the remaining scepticism is based on an opinion that is not true.
>
> The opinion that the Reviewer bases the negative opinion can be summarized as:
>
> *"Whether or not a reaction is assigned as Hallucination (Nonsense) is a bias of human, not reflection of the physical chance of success in the laboratory"*
>
> This is factually incorrect.
>
> We understand and appreciate that the Reviewer is a practitioner in the field, but we guess that the Reviewer is not a chemist.
>
> Does the Reviewer have access to chemists to show these examples?
>
> In Figure 1 of the paper, the SSR model proposes an ortho-amino benzoate that directly converts into a triazole ring. A PhD-level chemist can see that this does not belong to any known reaction class and that no reasonable set of conditions or reagents. A model achieving stellar top-1 accuracy proposed this transformation.
>
> Other examples can be found in Figure 9 & 11. In Figure 9, a nonaromatic ring is transformed into benzene. Any system showing such a transformation would be deemed incorrect by 100% of chemists.
>
> In other words, **the Hallucinations (Nonsense category) are not an opinion in the sense that almost surely there is no known synthetic approach to achieve those transformations.**
>
> (That's precisely the reason that human evaluation was used in Segler's Nature paper.)

---

> > ### Comment · Reviewer_qjSH · 2025-11-27
> >
> > I think you misunderstand what I mean.
> >
> > Of course I understand that hallucinated reactions are not feasible in reality. What I mean is that, to determine whether a model’s predicted reaction is truly feasible, we should rely on wet-lab experiments or on established, reliable chemical reaction databases, rather than on individual human judgment alone, which is inevitably prone to error.

---

> > > ### Author Response · Authors · 2025-11-27
> > > **Thank you**
> > >
> > > Thank you again for your prompt response.
> > >
> > > We agree that, in general, correctness cannot be asserted by humans. We do not claim that! We are happy to revise the manuscript to clarify that.
> > >
> > > However, **specifically for the Hallucinated (Nonsense) category (focus of our paper and title): yes it can.** You said yourself "hallucinated reactions are not feasible in reality".
> > >
> > > There is no significant disagreement between chemists when assigning this category. 100% of those predictions would be confirmed in the lab.
> > >
> > > Your opinion that "hallucinated (Nonsense) reactions are not feasible in reality" implies logically speaking that "testing whether a system hallucinates grossly incorrect reactions can be tested by asking humans instead of doing a lab experiment".
> > >
> > > (Please further note that using public databases of reactions does not allow to understand the hallucination rate of retrosynthesis systems. )

---

> > > > ### Author Response · Authors · 2025-12-03
> > > > **Ask**
> > > >
> > > > We would be grateful for answering our most recent comment. We have also uploaded the revised version.
> > > >
> > > > Thank you,
> > > > Authors

---

### Official Review · Reviewer_pA39 · 2025-11-01

**Soundness:** 3
**Presentation:** 3
**Contribution:** 4
**Rating:** 8
**Confidence:** 5

**Summary:**

This paper proposes a practical method to reduce hallucination rate and increase robustness of a retrosynthesis system by employing a combination of three complementary filters. The authors design each filter carefully and in a way that makes sense for the domain, show they are all helpful but complementary, and finally demonstrate that on their test set the combined filter removes all hallucinated predictions while maintaining a high solve rate.

**Strengths:**

**(S1)**: Hallucinated single-step predictions are a big issue in retrosynthesis systems. The approach designed by the authors is highly practical, and precisely targets the largest obstacle to real-world adoption for many existing retrosynthesis frameworks. Each of the three filters is designed in a way that builds on top of what worked in prior work, but extending those substantially.

**(S2)**: Authors set out to collect a dataset of chemist-annotated single-step predictions, which they then use to calibrate and evaluate their filters; this is precisely the right approach in my opinion. The dataset they generate is also fairly large given the context (and would be highly useful if released to the community).

**(S3)**: Experiments convincingly show that the filters are complementary, and using a combination of them is highly effective in increasing the robustness of the pipeline.

**Weaknesses:**

**(W1)**: There are some aspects of this work that are not clear to me:

- **(W1a)**: Authors explain that synthetic negative reactions are generated by applying random templates in both forward and backward direction. For the forward direction, this is standard; applying a random template to reactants from the dataset can be assumed to produce a negative reaction as long as the product differs from the one recorded in the data (this hinges on the assumption that a given set of reactants can only react to produce one potential product, which is not necessarily true due to missing conditions and reagents, but approximately this assumption would hold in most cases). However, for the backward direction, how to guarantee that the proposals are not valid ways of synthesizing the given product? Are the backward-generated synthetic negatives only accepted if they then fail round-trip with a forward model?

- **(W1b)**: Every score that is part of RP uses a different normalization (e.g. dividing by number of summed probabilities vs by the square root of that number). Is there an intuition behind this, or was this simply determined empirically to maximize the predictive power of each score?

- **(W1c)**: What does "reaction count within the candidate reaction’s coarse-grained cluster and fine-grained cluster" mean? Are the authors referring to former or latter? I suppose one could also read this as intersection of coarse-grained and fine-grained, but I assume the latter is a subset of the former.

- **(W1d)**: In Section 3.5, authors mention a "BART generator" is used as the single-step model, yet later it seems RootAligned was their default choice?

- **(W1e)**: Authors mention a subset of the chemist-annotated dataset of good/bad single-step model generations will be "released as a benchmark for the community". How many reactions/annotations are being released?

- **(W1f)**: In Section 5.2, authors mention the retrosynthetic paths were assigned a confidence score determined by the lowest-scoring reaction within them. However, doesn't this mean a human would have to score the routes generated in this experiment to obtain the confidence score?

---

**Other comments**

**(O1)**: It would be good to relate the results from this work to those from the recent RetroChimera paper [1], which also shows strong results based on ensembling. That work employs an ensemble of complementary single-step models, and shows this significantly improves both coverage and ranking of the proposed reactions. However, although hallucinated predictions are pushed down to lower ranks, and may potentially get truncated out when limiting the number of predictions to include in search, some incorrect reactions still remain (see Extended Data Figures 12-14 in [1]). RetroTrim mirrors these findings, showing ensembling is highly successful also for the case of reaction scorers.

**Nitpicks**

- The use of parentheses around citations is not consistent with common practice. Whenever citation appears as part of the sentence it should not be parenthesized, but if it appears as a remark outside of sentence it should be. There are cases of this throughout the manuscript, e.g. at the beginning of introduction or beginning of Section 2.1.

- Would be nice to also include which version of Pistachio was used (i.e. from which quarter).

- Some parts of equations, e.g. Equation 1, use long words without special formatting, which can make them look a bit unprofessional. I would consider using the `\texttt` command around words like `reaction` or `ref`.

**References**

[1] "Chemist-aligned retrosynthesis by ensembling diverse inductive bias models"

**Questions:**

See the "Weaknesses" section above for specific questions.

---

> ### Author Response · Authors · 2025-11-21
> **Thank you very much**
>
> Thank you very much for your thoughtful feedback and questions.
>
> **Response to Weaknesses**
>
> **W1a — Synthetic backward negatives**
> This is a very good point. Note that the issue you raise also applies to forward negatives: since the model does not see conditions, it is entirely plausible (though less likely) that a “negative” forward template could in fact correspond to a valid reaction under different conditions (e.g., thermal vs. kinetic control).
> Therefore, the same conceptual flaw exists for both forward and backward synthetic negatives, but in practice it does not prevent them from being useful training signals. We agree that filtering backward negatives with a forward model could further improve their quality. In our experiments, we validated that both forward and backward synthetic negatives are helpful, and we will mention this, as well as the potential improvement via forward-model filtering, in the camera-ready version.
>
> **W1b — Normalization choices in RP scores**
> The normalization schemes for the Reaction Prior (RP) components were selected empirically to maximize predictive performance on a set of 4,500 manually evaluated reactions. We will clarify this in the paper and add a short explanation that these choices were driven by empirical calibration rather than purely heuristic design.
>
> **W1c — “Reaction count within the candidate reaction’s coarse-grained cluster and fine-grained cluster”**
> Apologies for the lack of clarity. We will spell this out more explicitly in the final version. Concretely, we count how many unique reactions in the training set are marked as similar to the candidate under our similarity metric—either very similar (“fine-grained”) or somewhat similar (“coarse-grained”). The higher this count, the higher (after a log transform) the score assigned to the candidate reaction. We will update the text to make this definition precise.
>
> **W1d — BART vs. RootAligned**
> We used these terms somewhat interchangeably in the current draft. The single-step generator has a BART-based architecture and is trained on root-aligned SMILES. In the camera-ready version, we will consistently refer to this as a BART-based, root-aligned model and avoid any ambiguity.
>
> **W1e — Size of the released chemist-annotated subset**
> We plan to release a subset of 500 reactions from the chemist-annotated dataset. We will clearly state this number in the paper.
>
> **W1f — Path confidence score vs. human scoring**
> Yes, in this evaluation setup a human expert assessed every reaction in each proposed route. The route-level confidence score was then defined as the minimum of the reaction-level scores along that route. We will make this evaluation protocol more explicit in the main text.
>
> **Response to Other Comments**
>
> **O1 — RetroChimera**
> After the paper was submitted, we added RetroChimera as an additional baseline. Under our multi-step retrosynthesis evaluation and annotation protocol, RetroChimera’s hallucination rate is similar to that of the other baseline methods. We will include these results and a brief discussion in the camera-ready version.
>
> **Minor edits**
>
> We will:
>
> fix inconsistent citation formatting,
>
> state the exact Pistachio release used, and
>
> improve equation typesetting (e.g., using \texttt{} where appropriate).

---

### Official Review · Reviewer_M6L6 · 2025-11-01

**Soundness:** 3
**Presentation:** 3
**Contribution:** 3
**Rating:** 6
**Confidence:** 3

**Summary:**

This paper introduces RetroTrim, a trustworthy retrosynthesis framework designed to eliminate hallucinated reactions. The method integrates three complementary reaction scorers to prune invalid reactions dynamically during multi-step retrosynthetic search. The authors further propose a human expert annotation protocol, defining seven hallucination types and three severity levels. On 32 drug-like benchmark targets, RetroTrim is reported to completely remove hallucinated reactions, outperforming all baselines both in accuracy and the number of valid synthetic routes found.

**Strengths:**

* The paper focuses on the trustworthiness of synthetic route prediction in retrosynthesis and clearly illustrates the negative impact of hallucinated reactions on the reliability of computational synthesis planning.

* The proposed method mimics the multi-dimensional reasoning process of human chemists when evaluating reaction feasibility, combining both novelty and practical utility.

* The authors establish a well-structured expert annotation protocol, defining explicit error categories and severity levels, which are consistently recognized among annotators.

* The proposed method outperforms all existing baselines across every major metric and shows strong potential for direct integration into existing retrosynthesis platforms.

**Weaknesses:**

* The proposed method increases inference time by approximately 2.3× compared to baseline systems, which may limit its scalability in high-throughput drug discovery pipelines.

* The RRS scorer's reliance on existing reaction precedents could introduce bias against novel yet chemically valid reactions, thereby potentially suppressing creative synthetic pathways.

* The ensemble aggregation is based on a fixed weighted average, lacking adaptive or learnable optimization mechanisms that could further improve robustness.

* While the paper reports pruning efficiency, it does not provide an analysis of the trade-off between hallucination filtering strength and route diversity or completeness.

**Questions:**

* How sensitive is RetroTrim's performance to the choice of the ensemble threshold?

* Among the few reactions that were incorrectly filtered out ("false negatives"), what is their chemical nature or common pattern?

* Could the scoring ensemble be integrated end-to-end with the single-step retrosynthesis generator?

---

> ### Author Response · Authors · 2025-11-21
> **Thank you for your time and helpful feedback.**
>
> Thank you for your time and helpful feedback.
>
> **Response to Weaknesses**
>
> **Inference-time overhead**
> We believe there may be a misunderstanding regarding the reported 2.3× inference-time overhead. In our experiments, retrosynthesis searches with scorers were run for a fixed budget of 500 expansion steps, whereas baseline systems were run for 15 minutes. For single-step prediction, the scorer’s inference time is much shorter than the generator’s (the generator is roughly ten times slower). Under the same expansion-step budget as in the paper, adding the scorers makes inference only slightly longer, and we view this as a reasonable trade-off for substantially reducing hallucinations. We will clarify this setup more explicitly in the revision to avoid confusion.
>
> **Novelty bias of the RRS scorer**
> We agree that Worthwhile reactions should sometimes be retained, and that this is inherently a precision–recall trade-off. In our work, we tuned the threshold so that recall remains comparable to, or better than, competing solutions, while precision (i.e., hallucination rejection) is substantially higher. We will make this trade-off and the corresponding empirical evidence more explicit in the paper.
>
> **Fixed-weight ensemble vs. adaptive optimization**
> We experimented with learning the ensemble weights on a set of 4,500 reactions. However, this tuning set was not representative of the distribution encountered during multi-step search (where the number and nature of reactions grow exponentially by depth), and the learned weights actually degraded performance compared to a fixed-weight ensemble. This suggests that more sophisticated or adaptive schemes would need to carefully account for this distribution shift. We agree this is a promising direction and will highlight it more clearly as future work in the Conclusions.
>
> **Trade-off between hallucination filtering and route diversity**
> In this paper, we focused on the correctness of top-1 pathways as our primary objective. Empirically, we did not observe a noticeable decrease in the diversity of the remaining routes, but we did not perform a dedicated quantitative analysis of this trade-off. We will clarify this limitation in the text and outline a more systematic diversity/completeness study as future work.
>
> **Response to Questions**
>
> **Q1 — Sensitivity to the ensemble threshold**
> The ensemble threshold is indeed an important hyperparameter that needs tuning. Fortunately, this tuning is relatively cheap computationally, as it can be performed off-line on the evaluation dataset without re-running search. We will clarify our tuning protocol and add a brief sensitivity analysis in the camera ready version.
>
> **Q2 — Nature of filtered-out but valid reactions (“false negatives”)**
> This is a very insightful question. One likely source of false negatives is the failure to find a suitable precedent under our fixed similarity function. In other words, a chemically valid reaction may be penalized simply because it falls just outside our notion of “similar” in the training data. We will explicitly call out this failure mode and discuss it as an avenue for improving the similarity metric in future work.
>
> **Q3 — End-to-end integration with the single-step model**
> Yes, the scoring ensemble can be integrated end-to-end with the single-step retrosynthesis generator, and can also be used as a standalone reliability layer for single-step predictions. We will clarify this in the paper and briefly discuss such integration scenarios as a natural extension of our framework.

---

### Author Response · Authors · 2025-12-03
**Thank you**

We extend our thanks to the Area Chair and Reviewers. **We also respectfully draw attention to the updated evaluation from Reviewer qjSH27, who raised their score (2 $\rightarrow$ 4).**

**Hallucinations plague generative chemistry models just as they do LLMs.**

Our paper has high novelty: we provide the first dedicated solution to this issue, offering both a comprehensive benchmark and the first high-recall system to generate pathways without "Nonsense" outputs.

Figure 1 exemplifies why this work is necessary: **a model with stellar top-1 accuracy proposed a chemically impossible transformation (ortho-amino benzoate directly to a triazole ring)**. This proves that accuracy metrics alone can mask severe factuality issues.

Recognizing that the negative reviews stem from skepticism regarding the severity of hallucinations, our edits provide further evidence that solving this problem is a prerequisite for reliable generative chemistry. We describe the changes below.

---

> ### Author Response · Authors · 2025-12-03
> **A brief summary of changes**
>
> For convenience, we summarize briefly changes made to the paper. We have updated the manuscript to include RetroChimera performance results, which further validate the prevalence of hallucinations in retrosynthesis and the efficacy of our RetroTrim plausibility filtering. We significantly clarified the system architecture by updating Figure 2 and explicitly defining the relationship between RetroTrim, RootAligned, and the MetaScorer. Finally, we expanded the text to emphasize the indispensability of human expert analysis for identifying nonsensical reactions and addressed reviewer concerns regarding experimental methodology and scoring design.

---

### Meta-Review · Area_Chair_ro7A · 2026-01-06

**Summary:**

This paper proposes RetroTrim, a retrosynthesis framework aimed at improving trustworthiness by filtering hallucinated reactions using an ensemble of diverse reaction scorers (Reaction Prior, Reaction Center, Regioselectivity, and a forward scorer). The authors argue that hallucinated, chemically nonsensical steps are a major obstacle to deploying generative retrosynthesis systems in practice, and introduce a chemist-annotated evaluation protocol to assess reaction-level and route-level correctness on challenging drug-like targets.

Reviewers generally agreed that hallucinations are an important and underexplored problem in retrosynthesis, and several appreciated the practical motivation, the use of domain knowledge, and the effort invested in expert annotation. The rebuttal addressed many technical clarification requests and improved the exposition of the scoring components and experimental setup. However, significant concerns remain regarding evaluation and the reliance on human expert judgment as primary ground truth in an ML venue. These unresolved concerns, raised consistently by multiple reviewers and not satisfactorily addressed (checked by AC), ultimately motivate the rejection decision.

**Reviewer Concerns:**

Concerns that have been addressed satisfactorily:
- In response to concerns about implementation details and clarity raised by Reviewer pA39 and Reviewer 38oB: the authors clarified the construction of synthetic negatives, normalization choices in Reaction Prior components, the relationship between BART and RootAligned generators, and corrected ambiguities in figures and captions.
- In response to concerns about ensemble design raised by Reviewer M6L6: the authors explained why fixed-weight aggregation was chosen over learned weights and provided justification based on observed distribution shift during multi-step search.
- In response to concerns about trade-offs between hallucination filtering and recall raised by Reviewer 38oB: the authors acknowledged the precision–recall trade-off, clarified threshold tuning, and committed to discussing route diversity and completeness more explicitly in the revised manuscript.
- In response to reproducibility concerns raised by Reviewer 38oB: the authors agreed to add USPTO evaluations in a future revision and clarified dataset usage and release plans.

Concerns that have not been addressed satisfactorily:
- Evaluation validity and ground truth definition (Reviewer qjSH): the reviewer consistently argued that reaction model confidence scores (Reaction Prior, Reaction Center, Regioselectivity) are not principled indicators of chemical feasibility, and that human expert judgment—while valuable—is inherently subjective and insufficient as primary ground truth in an ML conference context. Despite extensive discussion, the authors did not provide objective, scalable alternatives (e.g., database matching or forward-model reconstruction) that would resolve this concern.
- Scope and strength of empirical evidence (Reviewer qjSH): the evaluation relies on a relatively small number of targets (32) and heavily on costly human annotation, limiting statistical confidence and generalizability. This concern was acknowledged but not resolved; in fact, previous retrosynthesis works, including GRASP [1] evaluated on 50 (which is larger than the authors' "first"claim) human-annotated targets. Results on USPTO were not provided either.

[1] Grasp: navigating retrosynthetic planning with goal driven policy

**Reviewer Scores:**

- Reviewer pA39: Initially positive (8); would likely maintain a high score, viewing the work as practically valuable despite limitations.
- Reviewer M6L6: Marginally positive (6); would likely maintain the score after clarification but not strongly advocate acceptance.
- Reviewer qjSH: Negative (2); would likely maintain the score, unconvinced about evaluation methodology and suitability for an ML venue.
- Reviewer 38oB: Negative (2); indicated openness to reconsideration pending revision (likely 4 as key evaluation concerns remain unresolved).

---

### Decision · Program_Chairs · 2026-01-26

Reject